# The p53–53BP1-Related Survival of A549 and H1299 Human Lung Cancer Cells after Multifractionated Radiotherapy Demonstrated Different Response to Additional Acute X-ray Exposure

**DOI:** 10.3390/ijms21093342

**Published:** 2020-05-08

**Authors:** Margarita Pustovalova, Lina Alhaddad, Nadezhda Smetanina, Anna Chigasova, Taisia Blokhina, Roman Chuprov-Netochin, Andreyan N. Osipov, Sergey Leonov

**Affiliations:** 1School of Biological and Medical Physics, Moscow Institute of Physics and Technology, 141700 Dolgoprudny, Moscow Region, Russia; lina-alhaddad@hotmail.com (L.A.); smetaninanm@gmail.com (N.S.); annagrekhova1@gmail.com (A.C.); tai2509@yandex.ru (T.B.); netochin@gmail.com (R.C.-N.); 2State Research Center-Burnasyan Federal Medical Biophysical Center of Federal Medical Biological Agency (SRC-FMBC), 123098 Moscow, Russia; 3Emanuel Institute for Biochemical Physics, Russian Academy of Sciences, 119334 Moscow, Russia; 4Semenov Institute of Chemical Physics, Russian Academy of Sciences, 119991 Moscow, Russia; 5Institute of Cell Biophysics, Russian Academy of Sciences, 142290 Pushchino, Moscow Region, Russia

**Keywords:** non-small cell lung cancer, ionizing radiation, radioresistance, DNA repair, p53

## Abstract

Radiation therapy is one of the main methods of treating patients with non-small cell lung cancer (NSCLC). However, the resistance of tumor cells to exposure remains the main factor that limits successful therapeutic outcome. To study the molecular/cellular mechanisms of increased resistance of NSCLC to ionizing radiation (IR) exposure, we compared A549 (p53 wild-type) and H1299 (p53-deficient) cells, the two NSCLC cell lines. Using fractionated X-ray irradiation of these cells at a total dose of 60 Gy, we obtained the survived populations and named them A549IR and H1299IR, respectively. Further characterization of these cells showed multiple alterations compared to parental NSCLC cells. The additional 2 Gy exposure led to significant changes in the kinetics of γH2AX and phosphorylated ataxia telangiectasia mutated (pATM) foci numbers in A549IR and H1299IR compared to parental NSCLC cells. Whereas A549, A549IR, and H1299 cells demonstrated clear two-component kinetics of DNA double-strand break (DSB) repair, H1299IR showed slower kinetics of γH2AX foci disappearance with the presence of around 50% of the foci 8 h post-IR. The character of H2AX phosphorylation in these cells was pATM-independent. A decrease of residual γH2AX/53BP1 foci number was observed in both A549IR and H1299IR compared to parental cells post-IR at extra doses of 2, 4, and 6 Gy. This process was accompanied with the changes in the proliferation, cell cycle, apoptosis, and the expression of ATP-binding cassette sub-family G member 2 (ABCG2, also designated as CDw338 and the breast cancer resistance protein (BCRP)) protein. Our study provides strong evidence that different DNA repair mechanisms are activated by multifraction radiotherapy (MFR), as well as single-dose IR, and that the enhanced cellular survival after MFR is reliant on both p53 and 53BP1 signaling along with non-homologous end-joining (NHEJ). Our results are of clinical significance as they can guide the choice of the most effective IR regimen by analyzing the expression status of the p53–53BP1 pathway in tumors and thereby maximize therapeutic benefits for the patients while minimizing collateral damage to normal tissue.

## 1. Introduction

Non-small cell lung cancer (NSCLC) is the most frequently diagnosed cancer worldwide [1]. It represents nearly 90% of all lung cancer diagnoses. More than one-half of patients with NSCLC are diagnosed with locally advanced (stage III) and advanced (stage IV) disease. The prognosis for lung cancer remains poor, with overall 5 year survival of 14% [2]. The role of curative-intent radiotherapy (RT) is well established in locally advanced and early stage NSCLC [3]. The proportion of patients with NSCLC with evidence for radiotherapy (RT), according to current estimates, ranges from 46% to 68% for primary patients and 64% to 75% for the entire cohort of patients with NSCLC. Moreover, the actual use of RT in the whole world is lower, in the range from 28% to 53%, with the largest differences between the actual and estimated use of radiotherapy for stage III NSCLC [4]. However, radioresistance and cancer recurrence are major obstacles for the long-term survival of patients undergoing RT [5].

Tumors may comprise subpopulations of cells with distinct genomic alterations and molecular signatures, a phenomenon termed intra-tumor heterogeneity [6]. Intra-tumor heterogeneity results in the ability of a tumor to harbor cells with differential levels of sensitivity to treatment [7]. Two models have been postulated to explain this phenomenon—the clonal evolution model and cancer stem cells (CSCs) model. The clonal evolution model proposes that cancer cells accumulate genetic changes due to genomic instability. These changes allow the cells to acquire advantageous characteristics and to be selected in a Darwinian-like evolutionary process [8]. The CSCs model proposes that, within a tumor, a small subpopulation of cells exhibit properties of self-renewal and plasticity, as well as the capability of reestablishing a heterogeneous tumor cell population [9]. Recently, the CSCs theory has offered a potential explanation for the relapse and resistance that occurs in many tumors [10]. CSCs has been associated with intrinsic determinants (DNA repair capability, reactive oxygen species (ROS) levels, cell cycle status, autophagy, apoptosis, regulation of survival pathway) and extrinsic determinants (the influence of hypoxic microenvironment) [5]. Radiation may selectively kill the relatively radiosensitive tumor cell populations, leaving the therapy-resistant CSCs alive, thus contributing to adaptive radioresistance via the selective repopulation from the surviving CSCs [11].

Exposure of cancer cells to fractionated ionizing radiation (IR) regimens can select a cancer subpopulation with modified cell characteristics in response to subsequent radiation exposure and affect tumor control probability [12]. This selection process is widely used as an experimental tool for obtaining radioresistant subpopulations of cancer cells. It helps to investigate their molecular response and guide the improvements in radiotherapy standards [13,14]. For locally advanced NSCLC, the standard treatment approach is conventionally fractionated (1.8–2.0 Gy/day) radiotherapy to a dose of approximately 60–66 Gy with concurrent, platinum-based chemotherapy [15,16]. Stereotactic body radiation therapy (SBRT) is a potentially effective technology to manage early stage NSCLC for inoperable patients and is often used for lung cancer patients with thoracic recurrences previously treated with conventional radiotherapy [17]. SBRT allows clinicians to administer high doses (commonly used doses are 48–50 Gy in 4–5 fractions and 50–60 Gy in 8–10 fractions) to accurately target the tumor [18]. However, both local and regional/distant recurrences still occur after SBRT [19]. In this study, we applied multiple fractions of subsequently increasing doses to obtain the populations of radioresistant cells. The combination of 10 fractions of 2 Gy, 4 fractions of 5 Gy, and 2 fractions of 10 Gy was not used in the clinic. However, this combination could serve as the representation of the conventional radiotherapy treatment with subsequent SBRT. The aim of the present study was to investigate the nature of NSCLC cell lines (p53 wild-type A549 and p53-null H1299) survived after multi-fractionated irradiations in total dose of 60 Gy. In the present study, we found that the H1299IR (p53-deficient) cells demonstrated a more prominent CSC-like phenotype in comparison to A549IR (p53-wild type) cells, also suggesting that, albeit the proliferation of parental cancer cells was p53-dependent, the proliferation of the cells surviving after multifraction radiotherapy (MFR) was p53-independent. Interestingly, in response to incrementing acute single-dose IR exposure, functional p53 conferred a significant dose-dependent decrease in the fraction of Ki67+ to both A549 and A549IR cells, whereas in the absence of functional p53, both parental and MFR-surviving cells acquired only subtle cell cycling decrease. Our data strongly support the notion that the presence of functional p53 resulted in durable IR-induced G1 and G2 arrests, whereas the absence of p53 led both parental and IR-resistant NSCLC cells to exit G0/G1 and to a more prominent arrest in G2/M phase. Moreover, our data drew attention to the efficient and p53-independent 53BP1 recruitment to damaged chromatin and 53BP1-mediated non-homologous end-joining (NHEJ), thus favoring either the survival advantage p53-null cells (higher proliferation/DNA replication and lower apoptosis) or induction of CSC-like phenotype observed in the present study. However, our current study demonstrated that 53BP1-related survival NSCLC cells after MFR was not associated with p53 functionality. Moreover, in our experiments, the enhanced 53BP1-mediated cell survival after MFR did not point to any particular type of cell cycle checkpoint response and could not be amply explained by its role in regulating checkpoint or pro-apoptotic signaling. The present data add value to the debated p53–MDR relationship in cancer by indicating that p53 status affects ABCG2 expression in cells surviving after MFR, and indicate a major role of p53-family proteins in conferring a stem-like cell phenotype and radioresistance to NSCLC cells that are associated with ABCG2 overexpression.

## 2. Results

### 2.1. Establishment and Phenotypic Characterization of A549IR and H1299IR Cells

Exponentially growing A549 and H1299 cells were irradiated with clinically relevant doses of IR (10 fractions of 2 Gy, 4 fractions of 5 Gy, and 2 fractions of 10 Gy). The majority of cells died after exposure to total dose of 60 Gy and the surviving cells demonstrated enlarged, flattened morphology. Untreated parental A549 and H1299 cells were cultured under the same conditions without irradiation. Radiation-surviving cells starting clonogenic growth were named as A549IR and H1299IR sublines. All experiments were conducted on cells that did not exceed 10 passages.

A549IR cells did not differ in their morphology compared to parental A549 (p53-wild type) cells. The H1299IR cells morphologically changed into a spindled or rounded shape in contrast to parental H1299 (p53-deficient) cells, which displayed typical polygonal, epithelial morphology. Moreover, the H1299IR cells demonstrated loss of cell–cell contacts and increased number of round viable cells with reduced adherence to the plastic (Figure 1b). These morphological changes suggested that a more prominent epithelial–mesenchymal transition (EMT)-like phenotype occurred in H1299IR (p53-deficient) in comparison to A549IR (p53-wild type) cells.

The ability of NSCLC cells not only to survive but also to retain reproductive potential after radiation underlies the problem of tumor recurrences seen in patients. Therefore, we tested the ability of every cell in the population of both the parental cells and their IR-surviving sublines to undergo “unlimited” division growing into a colony. The colony is defined as consisting of at least 50 cells. Clonogenic survival was assessed using a standard colony formation assay that measured the loss of reproductive integrity after exposure to IR. Clonogenic assay or colony formation assay is an in vitro cell survival assay of choice to determine cell reproductive death after chemo- or radiation treatment [20].

The A549IR cells demonstrated insignificantly higher plating efficiency compared to parental A549 (p53 wild-type) cells growing under normal physiological conditions (Figure 2a). In contrast, the H1299IR cells had more than seven times lower plating efficiency compared to parental H1299 (p53-deficient) cells (Figure 2b). Whereas the clonogenic cell potential of parental A549 cells (p53 wild-type) were lower compared with H1299 (p53 deficient) cells, the A549IR and H1299IR cell sublines survived after IR in a total dose of 60Gy, demonstrating quite the opposite clonogenic behavior. The obtained data suggested that functional p53 increased clonogenic survival of X-ray-resistant sublines derived from parental cells exposed to fractionated IR.

### 2.2. Proliferative Activity of A549IR and H1299IR Cells Decreased Compared to Parental Cells in a p53-Dependent Manner

Next, we investigated the impact of p53 on the proliferation of the sublines that survived after multifraction X-ray irradiation. We used Click-iT^TM^ EdU and Ki67 immuno-fluorescent staining as the two complimentary proliferation-related assays. The former, on the basis of incorporation of the thymidine analogue EdU into newly synthesized DNA, quantifies the dividing cells using high content analysis [21,22]. The Ki67 protein is present during all active phases of the cell cycle (G1, S, G2, and M) but is absent in resting (quiescent) cells (cell cycle phase G0). Significant increase of cellular content of Ki67 protein indicated cell progression through S phase of the cell cycle. However, this protein, in addition to be present in all DNA-replicating cells, was also detected in the G1 and G2/M population of cells that showed no evidence of EdU incorporation.

The level of EdU-positive cells was verified between parental and radiation-surviving cells. Albeit the fraction of proliferating parental A549 cells was lower compared to H1299 cells, the same fraction of radiation-surviving cells of both cell lines was statistically indistinguishable. The proliferation of A549IR cells grown both at high plating density of 6300 cells/cm^2^ and at low plating density (4700 cells/cm^2^) were not statistically significant (Figure 3a). H1299IR cells grown at both low and high plating densities also showed statistically significant (*p* = 0.02 and *p* = 0.01, respectively) reduction of proliferative activity compared to parental cells. These data may indicate that, although the proliferation of parental cells is p53-dependent, the proliferation of cells surviving after fractionated IR exposure is p53-independent.

To examine the proliferative activity after fractionated IR exposure, both A549IR and H1299IR along with their parental cells were exposed to three different single doses of acute X-ray irradiation. Non-irradiated A549IR and H1299IR as well as their parental cells were used as controls. Cells were harvested for Ki67 quantification by high content fluorescent analysis 24 h after each dose of irradiation.

As shown on Figure 4, although demonstrating trends similar to EdU incorporation, the percentage of Ki67+ cells did not differ significantly between non-irradiated parental and radiation-surviving cells of both sublines, thus implicating that their amount of DNA replicating cells and the cells in the growth–pre-replicative phase was not divergent in p53-dependent context. The EdU incorporation and Ki67 data suggested that functional p53 did not significantly influence the background proliferation level of radiation-surviving cells in contrast to parental cells.

A statistically significant IR dose-dependent decrease in the proportion of Ki67+ cells was observed in the population of parental A549 (p53 wild-type) cells exposed to single acute dose of 2, 4, and 6 Gy (*p* = 0.0075, *p* = 0.002, and *p* = 0.0035, respectively). The statistically significant decrease in the fraction of Ki67+ A549IR cells did mirror the one demonstrated by parental cells after single 2 and 6 Gy exposure (*p* = 0.03 and *p* = 0.0006, respectively), although it was not significant after 4 Gy IR exposure. In contrast, both parental H1299 (p53-deficient) and radiation-surviving H1299IR cells showed only statistically insignificant subtle decrease in the proportion of Ki67+ cells after single acute exposure at any IR dose in comparison to corresponding controls. We speculated that functional p53 might be essential for desired reduction of DNA-replicating cells and the cells in the growth–pre-replicative phase in response to acute IR exposure of both IR-resistant and parental cells.

### 2.3. Functional p53 Contributed to Increased G1 Arrest and Sustained G2 Arrest in Response to Single Doses of Acute X-ray Irradiation

To examine the relationship between cell cycle response and clonogenic survival after fractionated IR exposure, both A549IR and H1299IR along with their parental cells were exposed to three different single doses of acute X-ray irradiation. Cells were harvested for cell cycle analysis by flow cytometry 24 h after exposure. Non-irradiated A549IR and H1299IR as well as their parental cells were used as controls. Representative data are shown in Figure 5.

Cell cycle analysis showed the percentage of G1/G0 cells (Figure 5a) was slightly higher only in A549IR cells with functional p53 at 24 h after each single acute IR dose of 2, 4 and 6 Gy (82%, 80%, and 78%, respectively, versus 70% in control). The persistent smooth increase in G1 cells and reduction in cells entering S phase in both parental and IR-resistant cells in comparison to the unirradiated control (Figure 5b) revealed the influence of functional p53 on G1 arrest. The sustained reductions of S phase (−12% after 6 Gy) were coincident with persistent elevations in the percentage of cells in G2/M (+15% at the same dose), but only for parental A549 cells (Figure 5c). In contrast, without functional p53, S phase demonstrated smooth increase in parental, but not in IR-resistant subline with an overall cell accumulation that was twice as high after 4 Gy and 6 Gy doses of IR exposure of both H1299 and H1299IR cells (Figure 5f). Without p53, the percentage of G1/G0 cells (Figure 5e) dose-dependently decreased only in parental H1299 cells at 24 h after each single acute IR dose of 2, 4 and 6 Gy (65%, 58%, and 50%, respectively, versus 70% in control). While demonstrating lower background accumulation of G1/G0 cells, the H1299IR subline derived from p53-deficient H1299 cells possessed the same character of the G1/G0 changes as observed in A549IR subline derived from cells with functional p53. The fraction of cells in G2/M smoothly dose-dependently increased in both p53-deficient cell lines, approaching accumulations that were twice as high as those seen in functional p53 IR-resistant cells exposed to the same acute IR dose. Thus, the presence of functional p53 resulted in durable IR-induced G1 and G2 arrests, whereas the absence of p53 led both parental and IR-resistant cells to exiting of G0/G1 and more prominent arrest in G2/M phase.

### 2.4. Functional p53 Subsidized the Level of Apoptosis in Response to Single Doses of Acute X-ray Irradiation

Apoptosis is thought to play a crucial role in the resistance of cancer cells to a variety of anticancer therapies. Many studies suggest that induction of apoptosis correlates with cellular radiosensitivity. As shown in Figure 6, a substantially higher accumulation of G0/G1 cells of A549IR subline exposed to each single acute IR dose might provide suggestive evidence that p53-dependent apoptosis was also restored in cells surviving fractionated IR exposure.

To test this hypothesis, we investigated the effect of single dose acute IR on radiation-induced apoptosis of parental and irradiation-surviving cells using YO-PRO-1 and propidium iodide (PI) staining. The P2X7 receptor (P2X7R; an adenosine triphosphate (ATP)-gated, non-selective cation channel) was shown to be activated during apoptosis, leading to conversion of a non-selective cation channel to a cytolytic pore [23]. Certain dyes, such as the green fluorescent YO-PRO-1 dye, can enter these pores, whereas other dyes, such as PI, cannot. Thus, YO-PRO-1 can serve as an early marker of apoptotic cell death [24].

As expected, functional p53 dose-dependently increased the proportion of apoptotic cells both in parental and IR-surviving A549 cells after exposure to 2–6 Gy (Figure 6a). Of note, the A549IR cells showed significantly higher increase of the proportion of apoptotic cells in both the control group and after acute single-dose IR exposure compared to parental cells. In contrast, in the absence of p53, the proportion of apoptotic H1299IR cells decreased following irradiation, and reaching the statistically significant value only after 4 Gy exposure compared to unirradiated cells. Even though the proportion of apoptotic cells in the parental H1299 population decreased after 2 and 4 Gy exposure, at the 6 Gy dose, the apoptotic population significantly increased compared to unirradiated cells (Figure 6b). These data suggested that functional p53 sensitized both parental and IR-surviving cells to apoptosis in response to acute single doses IR. In contrary, the p53 deficiency led parental cells to prominent apoptotic response only at the highest dose of acute single dose IR, and rendered reduced apoptotic reaction in IR-surviving cells at any single acute dose. It was tempting to speculate that other modes of cell death (such as mitotic catastrophe or senescence) might have been engaged, responding to single acute dose of X-rays in IR-surviving cells in absence of p53.

### 2.5. The p53 Status Influenced the DNA Repair Capacity of Parental Cells and Their IR-Surviving Sublines in Response to Acute Single IR Exposure

DNA double-strand breaks (DSBs) are the most lethal DNA lesions induced after the exposure of cells to IR. DSB signaling and repair is crucial to preserve genomic integrity and maintain cellular homeostasis. Distinct γH2AX foci colocalized with phosphorylated ataxia telangiectasia mutated (pATM) represents a DSB-specific histone code of ongoing DNA repair processes. We aimed to answer the question whether p53 status affects assembly of localized DNA damage recognition and repair factor, thus promoting cancer cell recovery and survival after multifractionated IR.

Therefore, we performed a comparative analysis of γH2AX (DNA DSBs marker) and pATM foci localization in parental NSCLC cells and their IR-surviving sublines in response to single acute 2 Gy X-ray exposure. DNA DSBs detected by immunofluorescent staining for γH2AX and pATM foci were measured at various time points post-IR. Dynamics of each DSB marker in both parental and IR-resistant cells without IR exposure were used as controls. In the presence of wild-type p53, the kinetics of γH2AX (Figure 7a) and pATM (Figure 7b) foci formation in A549IR cells did not significantly differ from that of parental cells. By 8 h post-IR exposure, the number of γH2AX foci declined down to around 13% (A549IR cells) and around 25% (A549 cells). It accompanied by parallel respective reduction in number of pATM foci to approximately 30% (A549IR cells) and 37% (A549 cells) of their maximum values at 0.5 h. Accordingly, both DNA DSB markers congruently decreased to almost control level, indicating that most of the DSBs generated by a single dose of acute X-ray irradiation were repaired within 24 h.

In absence of p53, the kinetics of γH2AX and pATM foci formation in irradiation-surviving H1299IR cells significantly differed from that of parental cells. Indeed, the number of γH2AX (Figure 7c) and pATM (Figure 7d) foci was significantly (*p* = 0.02) higher in parental H1299 cells compared to H1299IR cells (45 vs. 25 and 35 vs. 15 foci/nucleus, respectively) at 30 min after acute exposure to single dose of 2 Gy. In addition, the parental H1299 cells demonstrated a significant sharp spike in the number of γH2AX and pATM foci (from 45 to 70 and from 34 to 46 foci/nucleus, respectively), with the maximum values observed at 1 h post-IR. Within the same period, the H1299IR cells possessed the modest change of γH2AX foci number. By 8 h post-IR exposure, the number of γH2AX foci declined down to around 20% (H1299 cells) and 50% (H1299IR cells) with parallel respective reduction in number of pATM foci to around 57% (H1299 cells) and 67% (H1299IR cells) of their maximum values at 1 h. Of note, the very subtle kinetic of pATM foci number was observed in irradiation-surviving H1299IR cells with almost complete return to the background values of controls by 8 h post-IR.

These data suggested that the p53 status affected early kinetics ofDNA DSBs repair in response to acute single-dose IR. The significantly higher number of γH2AX and pATM foci in p53-deficient cells compared to functional p53 cells may have been linked to either ineffective (incomplete) DNA DSB repair, or representing the impairment of DSB-specific histone code for sensing and assembling with DNA repair factor(s) during early phase of DNA DSB response (within 8 h post-IR).

### 2.6. The Impact of p53 on Residual DNA Repair Foci in Parental Cells and Their IR-Surviving Sublines in Response to Acute Single Doses of IR Exposure

The p53-binding protein 1 (53BP1) is an important regulator of the cellular response to DSBs that promotes non-homologous end-joining (NHEJ) of distal DNA ends while preventing homologous recombination (HR). The residual DNA repair foci observed in cells 24 h post-IR are useful markers for cellular radiosensitivity. Therefore, we analyzed the amount of residual γH2AX and p53-binding protein-1 (53BP1) foci in relation to p53 status of both parental and their IR-resistant sublines at 24 h after exposure to single acute doses of IR.

In the presence of functional p53, the number of residual γH2AX foci did not differ between parental and A549IR cells (Figure 8a) exposed to any single acute IR dose, whereas the number of residual 53BP1 foci was significantly lower in A549IR cells exposed to 6 Gy IR only compared to parental cells (Figure 8c). Nonetheless, the number of co-localized γH2AX/53BP1 foci in A549IR was significantly lower than in parental A549 cells exposed to 2 and 6 Gy (Figure 8e).

In contrast, in the absence of p53, the number of residual γH2AX foci in H1299IR cells was significantly lower than in parental cells after exposure to 4 and 6 Gy IR. The numbers of residual 53BP1 foci and co-localized γH2AX/53BP1 foci in the IR-surviving H1299IR subline were significantly lower than those of parental H1299 cells after exposure to any single acute IR dose. Of note, even in the absence of p53, both background and acute single-dose IR-induced levels of residual IRIF (γH2AX, 53BP1, and co-localized foci) in parental H1299 cells were significantly higher than the ones in parental A549 cells carrying functional p53. These data might indicate efficient and p53-independent 53BP1 recruitment to damaged chromatin and 53BP1-mediated NHEJ, thus favoring either the survival advantage p53-null cells (higher proliferation/DNA replication and lower apoptosis) or induction of the EMT-like phenotype observed in the present study.

### 2.7. Expression of ABCG2 in IR-Surviving Sublines and Their Parental Cells Depended on p53 Status in Response to Acute Single Doses of IR Exposure

The expression of *ABC* genes, such as *ABCG2* (ATP-binding cassette sub-family G member 2 protein - ABCG2, also designated as CDw338 and breast cancer resistance protein, BCRP), *ABCB1* (MDR1), and DNA repair gene *ERCC1*, were found to be significant predictors of poor prognosis for patients with advanced tumors [25,26]. Among ABC transporters, the expression of *ABCG2* is associated with a shorter survival in patients with advanced NSCLC when treated with cisplatin-based chemotherapy [25]. The ABCG2 is a suggested molecular determinant of the side population (SP) phenotype, and the expression of *ABCG2* mRNA was markedly higher in SP for all lung cancer cell lines analyzed [27].

Because our IR-surviving sublines demonstrated the EMT-like phenotype somewhat similar to the SP phenotype, we analyzed the expression of ABCG2 in parental cells and their IR-resistant sublines before and after acute single doses of IR exposure by in-cell ELISA. Due to the lower plating efficiency (PE) and proliferation rate of H1299IR cells compared to parental H1299 cells, the resulting signal was normalized to the number of cells per well.

As shown in Figure 9, the level of ABCG2 expression was significantly higher in parental A549 cells post-IR at single doses of 2, 4, and 6 Gy compared to unirradiated cells (*p* = 0.027, *p* = 0.01, and *p* = 0.045, respectively). Even though the ABCG2 expression in unirradiated A549IR cells was higher than in parental cells, the IR-surviving subline did not show any significant change in the level of ABCG2 expression compared to unirradiated cells (Figure 9a) in response to acute exposure to any dose of IR. In contrast, without acute single dose irradiation, the ABCG2 expression was higher in H1299IR than in parental H1299 cells (Figure 9b). The exposure of H1299IR cells to a single dose of 2 Gy led to dramatic (more than threefold) increase of ABCG2 expression over unirradiated cells, whereas at single acute doses of 4 and 6 Gy, the expression was downregulated compared to unirradiated cells. Of note, the ABCG2 expression in irradiated H1299IR cells remained significantly (*p* = 0.003, *p* = 0.018, and *p* = 0.007, at doses 2, 4 and 6 Gy, respectively) higher than in parental cells exposed to any single IR dose. These data might indicate that the p53 status of cells affects ABCG2 expression in cells surviving after fractionated IR exposure, pointing to a major role of p53-family proteins in conferring a stem-like cell phenotype and radioresistance of NSCLC cells that is associated with *ABCG2* overexpression.

## 3. Discussion

Multifractionated radiation (MFR), comprising the application of multiple fractions over a period of days or weeks, is one of the essential approaches of cancer treatment. Several different irradiation (IR) schedules with or without chemotherapy are presently explored in the clinical practice of NSCLC therapy. In many cases of tumor recurrence after primary radiotherapy, patients can rarely be treated again with the same radiation regimen because the tumor might acquire radioresistance [28]. The proposed processes contributing to enhanced cellular tolerance to MFR included cell cycle checkpoint responses and activation of pro-survival signaling pathways. In addition, it has been shown that radioresistance of cancer cells after multiple fractions of IR exposure can be linked to the induction of epithelial–mesenchymal transition (EMT), which is defined as the loss of epithelial cell characteristics, such as E-cadherin, and the gain of mesenchymal cell characteristics, such as N-cadherin, vimentin, snail, and twist [29,30]. Emerging evidence suggests that mutations in *TP53* are mechanistically linked with the regulation of EMT/MET equilibrium. The tumor suppressor protein p53 functions as the “guardian of the genome” by inducing cell cycle arrest, apoptosis, and senescence in response to oncogene activation, DNA damage, and other stress signals. p53 functions are lost in cancer cells due to mutations of the *TP53* gene or by inactivation of the p53 signal transduction pathway. The incidence of p53 mutation in adenocarcinoma is 45%–70% [31]. The majority of the mutations result in the overexpression of p53 protein with acquired oncogenic properties, such as invasion, metastasis, increased proliferation, and cell survival. Overexpression of p53 has been linked to poor survival in patients with endometrial carcinoma after adjuvant radiotherapy, suggesting its possible role in tumor radioresistance [32]. Nonetheless, the way in which human cells repair DNA damage in response to MFR remains largely unknown.

Here, we applied clinically relevant multiple fractions of increasing doses of X-ray irradiation to obtain and characterize irradiation-surviving populations of two NSCLC cell lines—A549 (wild type p53) and H1299 (p53 null), and to investigate molecular mechanisms underlying their survival and radioresistance in relation to their p53 status. Among irradiation-surviving cell sublines, we found that the H1299IR (p53-deficient) cells demonstrated a more prominent epithelial–mesenchymal transition (EMT)-like phenotype in comparison to A549IR (p53-wild type) cells. However, this assumption awaits our separate ongoing investigation.

Next, we evaluated the ability of NSCLC cells to retain reproductive potential after MFR, which underlies the problem of tumor recurrences seen in patients. We evaluated plating efficiency (PE) of parental and irradiation-surviving NSCLC cells. PE refers to the ability of plated cells to give rise to colonies. The A549IR cells demonstrated insignificant PE twice as high as parental A549 (p53-wild type) cells growing under normal physiological conditions (Figure 2a). In contrast, the H1299IR cells had more than seven times lower PE compared to parental H1299 (p53-deficient) cells (Figure 2b). Our data suggested that functional p53 increased clonogenic survival of IR-surviving sublines derived from parental NSCLC cells exposed to MFR, thus corroborating previous findings on prostate carcinoma cells in response to the MFR regimen [33].

In our study, the smaller size of H1299IR cell colonies (not shown) can be attributed to the lower S phase proliferation rate compared to parental cells. Indeed, we demonstrated that the level of EdU-positive (the fraction of proliferating cells) H1299IR cells was significantly lower than in parental cells (Figure 3b). Nonetheless, the proliferating fraction of parental A549 (functional p53) cells was significantly lower than H1299 (p53-null) cells, whereas the same fraction of IR-surviving cells of both cell lines was statistically indistinguishable. This led us to suggest that, although the proliferation of parental cancer cells was p53-dependent, the proliferation of the cells surviving after MFR exposure was p53-independent.

At the same time, we demonstrated that the fraction of cycling cells (Ki67+) of both parental cells and their IR-surviving sublines were almost similar (Figure 4). Interestingly, in response to incrementing acute single dose IR exposure, functional p53 conferred a significant dose-dependent decrease in the fraction of Ki67+ to both A549 and A549IR cells, whereas in the absence of functional p53, both parental and IR-surviving cells acquired only subtle cell cycling decrease. We speculated that functional p53 might be essential for the desired reduction of DNA-replicating cells and the cells in the growth–pre-replicative phase in response to acute IR exposure of both IR-surviving and parental cells.

Many studies suggest that DNA DSB repair is more efficient in cells that have survived repeated X-ray irradiation [34]. The histone variant H2AX phosphorylated at DSB sites by several protein kinases (ATM, ATR, and DNA-PK) called γH2AX is the most widely used marker of DNA DSBs [35,36,37]. ATM (ataxia telangiectasia mutated) is a major kinase that phosphorylates H2AX in response to IR-induced DNA damage [38,39]. It is also involved in multiple pathways including DNA repair checkpoint activation, cell-cycle control regulation, apoptosis, senescence, and alterations in chromatin structure, transcription, and pre-mRNA splicing [40]. In our study, clear two-component kinetics of DNA DSB repair after acute single-dose IR exposure of both parental and IR-surviving cells indicated the p53-independent kinetics of γH2AX foci formation and degradation. However, in the absence of functional p53, the rate of DSB repair seemed to be low, as the H1299IR cells still demonstrated the presence of around 50% of γH2AX foci at 8 h post-IR (Figure 7). In addition, low activation of pATM kinase in H1299IR cells may indicate the ATM-independent character of H2AX phosphorylation at early time points post-IR. Indeed, H2AX can be phosphorylated by ATR in the presence of single-stranded DNA and stalled DNA replication forks during S phase of the cell cycle. This is consistent with our results for cell cycle distribution of H1299IR cells after additional X-ray exposure—a decrease in the fraction of S and an increase in G2/M cells (Figure 5). Of note, the fraction of cells in G2/M smoothly IR dose-dependently increased in both p53-deficient cell lines, approaching the accumulations twice as high as those seen in functional p53 IR-surviving cells exposed at the same acute IR dose.

Xu et al. showed that the ATM-independent G2/M checkpoint occurs several hours post-IR and represents the accumulation of cells that had been in earlier phases of the cell cycle at the time of IR exposure [41]. This late G2/M accumulation suggests enrichment of cells lacking the IR-induced S phase checkpoint. Indeed, the decrease in S phase fraction of H1299IR cells observed in our study (Figure 5) might reflect the inability of these cells to activate intra-S checkpoint post-IR in absence of functional p53. The ATM–Chk2–p53 pathway became activated in response to DNA damage, being involved in regulation of G1 cell cycle arrest and apoptosis. The G1 checkpoint is defective in most cancer cells, commonly due to mutations/alterations of p53. In G0/G1 cells, around 80% of DSBs were repaired rapidly in the c-NHEJ pathway [42]. Similar to the results obtained by Li [43] for radioresistant A549 cells, we showed a statistically significant decrease in G1 phase and increase in the S and G2/M phases for IR-surviving H1299IR cells compared to parental H1299 cells, but not for A549IR cells. Thus, our data strongly support the notion that the presence of functional p53 resulted in durable IR-induced G1 and G2 arrests, whereas the absence of p53 led both parental and IR-surviving NSCLC cells to exiting of G0/G1 and more prominent arrest in the G2/M phase.

Löbrich and Jeggo demonstrated that G2 cells, similar to G1, show two-component DNA DSB repair kinetics, although the G2 process is slower than its G1 counterpart and can result in a high number of the residual γH2AX foci [44]. To evaluate the number of residual IRIF, we evaluated the number of another prominent DNA DSB marker, tumor suppressor p53 binding protein 1 (53BP1), which co-localizes with γH2AX foci and serves as a promoter of the NHEJ DSB repair pathway. Residual γH2AX/53BP1 foci are widely used for biodosimetry of IR exposure and may represent unrepaired or misrepaired DNA DSBs, chromatin alterations, apoptosis, and checkpoint signaling [45,46]. Our study demonstrated that, although showing linear dose-response in all cell populations, the number of co-localized γH2AX /53BP1 foci in A549IR was significantly lower than in parental A549 cells exposed to doses of 2 and 6 Gy (Figure 5e). Accordingly, the number of residual IRIF in the IR-surviving H1299IR subline was significantly lower than in parental H1299 cells after exposure to any single acute IR dose. In contrast, in absence of p53, both background and IR-induced levels of residual IRIF in parental H1299 cells were significantly higher than the ones in parental A549 cells carrying functional p53 (Figure 5e,f). These data might point out the efficient and p53-independent 53BP1 recruitment to damaged chromatin and 53BP1-mediated NHEJ, thus favoring either the survival advantage of p53-null cells (higher proliferation/DNA replication and lower apoptosis) or induction of EMT-like phenotype observed in the present study.

Besides its function in DNA repair, 53BP1 modulates radiation-induced cell cycle checkpoint responses and apoptosis through its interactions with the tumor suppressor p53 and the ubiquitin-specific protease USP28 [47,48]. Recent data have indicated that shielding of DNA ends against resection and p53 stabilization can be achieved by 53BP1 compartmentalization at DNA damage sites through phase separation [49]. Our current study demonstrated that 53BP1-related survival of NSCLC cells after MFR was not associated with p53 functionality. Moreover, in our experiments, the enhanced 53BP1-mediated cell survival after MFR did not point to any particular type of cell cycle checkpoint response and could not be amply explained by its role in regulating checkpoint or pro-apoptotic signaling. In our study, the number of colonies with an enhanced survival rate after MFR represented only a fraction of the original cells (conventionally 15%–60%). Therefore, one could not leave out the risk that any MFR -induced cell cycle change within this subpopulation might have been omitted in the bulk analysis.

Lack of apoptosis is a trait of radioresistant cancer cells [50]. In our study, functional p53 dose-dependently increased the proportion of apoptotic cells both in parental and IR-surviving A549 cells after exposure to 2-6 Gy (Figure 6a). In contrast, in the absence of p53, the proportion of apoptotic H1299IR cells decreased following irradiation, reaching the statistically significant value only after 4 Gy exposure compared to unirradiated cells (Figure 6b). Our present data suggested that functional p53 sensitized both parental and IR-resistant cells to apoptosis in response to acute single-dose IR. On the contrary, the p53 deficiency led parental cells to prominent apoptotic response only at the highest dose of acute single-dose IR, and rendered reduced apoptotic reaction in IR-resistant cells at any single acute dose. It was tempting to speculate that the p53 deficiency might have engaged other modes of cell death (such as mitotic catastrophe or senescence) in response to a single acute dose of X-rays in IR-surviving cells. Alternatively, our results might suggest that in the absence of p53, the DNA DSBs may be repaired through a relatively slow but more precise HR pathway, which contributes to the evasion of apoptosis and radioresistance.

A possible link was suggested between expression of p53 and CSCs formation and poor prognosis [51]. Mutations in *p53* were shown to allow stem-like characteristics in breast and lung cancers [52]. Among ABC transporters, the expression of *ABCG2* is associated with a shorter survival in patients with advanced NSCLC when treated with chemo- [25] and chemoradiotherapy [53]. *ABCG2* is one of the promising markers for CSCs identification and is known to contribute to multidrug resistance (MDR) in cancer chemotherapy [54]. The ABCG2 is a suggested molecular determinant of the side population (SP) phenotype, and the expression of *ABCG2* mRNA was markedly higher in SP for all lung cancer cell lines analyzed. Because our IR-surviving sublines demonstrated the EMT-like phenotype somewhat similar to the SP phenotype, we investigated whether MFR exposure augmented the expression of ABCG2 protein in relation to p53 status, thus indicating the changes in the amount of CSC-like populations in IR-surviving NSCLC cells. Even though the ABCG2 expression in unirradiated A549IR cells was higher than in parental cells, the IR-surviving subline did not show any significant change in the level of ABCG2 expression compared to unirradiated cells (Figure 9a) in response to acute exposure to any dose of IR. In contrast, the background ABCG2 expression was higher in H1299IR than in parental H1299 cells (Figure 9b), and the IR exposure of the former to a single dose led to dramatic (more than threefold at 2 Gy) increase of ABCG2 expression, albeit it was downregulated at single acute doses of 4 and 6 Gy compared to unirradiated cells. Of note, the ABCG2 expression in irradiated H1299IR cells remained significantly (*p* = 0.003, *p* = 0.018, and *p* = 0.007, at 2, 4 and 6 Gy, respectively) higher than in parental cells exposed to any single IR dose. It is known that ABCG2 is normally expressed in hematopoietic stem cells with the highest levels in the primitive bone marrow stem cell populations, followed by a sharp reduction in response to stem cell differentiation. This suggests a possible dual role of ABCG2 in maintaining human pluripotent stem cells in an undifferentiated state and in protecting these stem cells from xenobiotics or other toxins in vivo [55]. The same pattern was shown for CSCs reaction to IR exposure, which led to an increase in the proportion of CSCs in tumor through dedifferentiation of cancer cells into CSCs [56]. Another explanation of CSCs enrichment suggests different radiosensitivity of cancer cells and CSCs with subsequent apoptosis of cancer cells after IR exposure [56]. However, we did not observe an increase in the proportion of apoptotic cells in the H1299IR population, albeit not excluding the other forms of cell death (e.g., non-protecting autophagy). Further increase of the additional dose of IR exposure caused the reduction of ABCG2 signal in H1299IR cells. Our data were consistent with the assumption of tumor repopulation due to differentiation of CSCs and may point to a role of ABCG2 in autophagy regulation in these progenitor cells. Although this remains to be tested, our data support the view [57] that ABCG2 may protect CSCs against a variety of microenvironmental stressors, adding to the inherent resistance of these cells to both unfavorable milieus and standard anticancer (e.g., MFR) regimens. In this regard, our data add value to the debated p53–MDR relationship in cancer by indicating that p53 status affects ABCG2 expression in cells surviving after MFR and point to a major role of p53 in conferring a stem-like cell phenotype and radioresistance to NSCLC cells that is associated with ABCG2 overexpression.

From a therapeutic point of view, the pivotal role of p53 and 53BP1 in the adaptive response to MFR assumes that pharmacological targeting of the p53–53BP1 pathway can potentially amplify the MFR efficacy. Developing drugs that restore wild-type (WT) structure and function to p53 mutants is considered a very topical and demanding direction in recent anti-cancer molecular pharmacology. Indeed, over the past decades, several compounds have been reported to reactivate mutant p53; however, all but one (APR-246/PRIMA-1) have failed to progress to clinical development, despite previous strategies for mutant p53 reactivation having been thoroughly reviewed [58]. However, clinical evidence that this compound reactivates mutant p53 is still pending because the published phase I dose escalation study did not reveal this information [59]. Currently testing in clinical trials diminishing RNF8/RNF168 activity with clinically approved proteasome inhibitors (e.g., bortezomib) and multiple BET inhibitors can ameliorate the 53BP1 chromatin recruitment [60].

In summary, our study provides strong evidence that different DNA repair mechanisms are activated by MFR, as well as single-dose IR, and that the enhanced cellular survival after MFR is reliant on both p53 and 53BP1 signaling along with NHEJ. Our results are of clinical significance as they can guide the choice of the most effective IR regimen by analyzing the expression status of the p53–53BP1 pathway in tumors and thereby maximize therapeutic benefits for the patients while minimizing collateral damage to normal tissue. On the other hand, different adjuvant chemotherapy based on proteasome and/or BET inhibitors targeting this pathway have good potential to significantly amplifying the efficacy of MFR. In terms of receiving substantial clinical validation, such approaches eventually provide clinicians with personalized tumor genotype-associated radiotherapy, having improved therapeutic outcomes.

## 4. Materials and Methods

### 4.1. Cell Culture

A549 cell line was obtained from ATCC and cultured in DMEM (Gibco, Thermo Fisher Scientific, Waltham, MA, USA) containing 10% FBS, L-glutamine, and 1% penicillin/streptomycin (Sigma-Aldrich, St. Louis, MO, USA). H1299 cell line was obtained from ATCC and cultured in RPMI-1640 (Gibco, Fisher Scientific, Waltham, MA, USA) containing 10% FBS, L-glutamine, and 1% penicillin/streptomycin (Sigma-Aldrich, St. Louis, MO, USA). Cells were maintained in a humidified 5% CO_2_ environment at 37 °C.

### 4.2. Irradiation

Exponentially growing A549 and H1299 cells were irradiated using 200 kV X-rays RUB RUST-M1 X-irradiator (JSC “Ruselectronics”, Moscow, Russia) at the dose rate of 0.85 Gy/min (2.5 mA, 1.5 mm Al filter) at room temperature. A total dose of 60 Gy was divided into several doses as follows: 2 Gy of 10 fractions, 5 Gy of 4 fractions, and 10 Gy of 2 fractions. Cells were incubated for up to 3–4 days between fractionated doses of 5 and 10 Gy as a recovery period. After the last exposure, cells were maintained in normal growth conditions for 3 weeks to recover.

### 4.3. Assessment of Plating Efficiency

Exponentially growing cells were harvested by trypsin, plated in 6 mm Petri dishes, and incubated for 14 days at 37 °C. After that, cells were fixed with 99.5% methanol and stained with Giemsa dye. Colonies ≥ 50 cells were counted. Small colonies were counted under light microscope. Plating efficiency is the ratio of the number of colonies to the number of cells seeded.

### 4.4. EdU Cell Proliferation Assay

The Click-iT^TM^ EdU cell proliferation kit (Invitrogen, Thermo Fisher Scientific, Waltham, MA, USA) was used for cell proliferation assessment. Cells were seeded at concentrations of 1.5 × 10^3^ and 2 × 10^3^ cells/0.32 cm^2^ into a 96-well plate and incubated for 3 days. Then, 10 mM Stock Solution of EdU diluted in phosphate-buffered saline (PBS) at 1:1000 was added to cell cultures and incubated for 2.5 h at 5% CO_2_ humidified at 37 °C. After that, cells were fixed in 2% paraformaldehyde (PFA) at room temperature. Cell nuclei were stained with 40 μM Hoechst 33342 (Thermo Fisher Scientific, Waltham, MA, USA), and EdU-labeled cells were kept from direct light at 4 °C overnight. Following two rinses with PBS on the next day, Reaction Mix, containing dH_2_O, 50 mM CuSo4, 100 mM Tris (pH 8.5), 10% Triton, 1 M ascorbic acid, and 1 mM Alexa Flour azide (iFlour 488), was added to each well and incubated at room temperature in the dark for 1 h. Then, cells were rinsed several times in PBS. Cells were viewed and imaged using fluorescence microscopy (ImageXpress Micro XL, Molecular Devices LLC, San Jose, CA, USA).

### 4.5. Immunofluorescence Staining

Cells were seeded at the density of 6 × 10^3^ cells/cm^2^ onto coverslips (SPL Lifesciences, Gyeonggi-do, South Korea) placed inside 35 mm Petri dishes (Corning, NY, USA) 48 h before irradiation and fixed 0.5–24 h after irradiation in 4% paraformaldehyde in PBS (pH 7.4) for 15 min at room temperature, followed by two rinses in PBS and permeabilization in 0.3% Triton-X 100 (in PBS, pH 7.4) supplemented with 2% bovine serum albumin (BSA) to block non-specific antibody binding. Cells were then incubated for 1 hour at room temperature with primary rabbit monoclonal antibody against γH2AX (dilution 1:200, clone EP854(2)Y, Merck-Millipore, Burlington, VT, USA), primary mouse monoclonal antibody against phosphorylated ATM protein (dilution 1:200, clone 10H11.E12, Merck Millipore, Burlington, VT, USA), primary mouse monoclonal antibody against 53BP1 protein (dilution 1:200, clone BP18, Merck-Millipore, Burlington, VT, USA), or primary mouse monoclonal antibody against Ki67 protein (dilution 1:400, clone KiS5, Merck-Millipore, Burlington, VT, USA), which were diluted in PBS with 1% BSA. After several rinses with PBS, cells were incubated for 1 h with secondary antibodies IgG (H+L) goat anti-mouse (Alexa Fluor 488 conjugated, dilution 1:300; Merck-Millipore, Burlington, VT, USA), goat anti-rabbit (Rhodamine conjugated, dilution 1:300; Merck Millipore, Burlington, VT, USA), and goat anti-mouse (Alexa Fluor 555 conjugated, dilution 1:500; Merck-Millipore, Burlington, VT, USA) diluted in PBS (pH 7.4) with 1% BSA. Coverslips were then rinsed several times with PBS and mounted on microscope slides with ProLong Gold medium (Life Technologies, Carlsbad, SA, USA) with DAPI.Cells in coverslips were imaged using Nikon Eclipse Ni-U microscope (Nikon, Tokyo, Japan) equipped with a high definition camera ProgResMFcool (Jenoptik AG, Jena, Germany). Filter sets used were UV-2E/C (340–380 nm excitation and 435–485 nm emission), B-2E/C (465–495 nm excitation and 515–555 nm emission), and Y-2E/C (540–580 nm excitation and 600–660 nm emission). A total of 300–400 cells were imaged for each data point. Foci were counted by manual scoring.

### 4.6. Cell Cycle Analysis

Exponentially growing cells were irradiated (2, 4, or 6 Gy) and incubated for 24 hours post-radiation. Cells were then harvested and 1 × 10^6^ cells per sample were resuspended in ice-cold PBS. Cells were fixed in ice-cold 70% ethanol for 30 min at 4 °C and then kept at −20 °C until analyzed. Before analysis, specimens were washed twice with PBS, and resuspended in 0.5 mg/mL PI containing 50 μl of 100 μg/mL RNase for 30 min in the dark. Cells were analyzed by flow cytometry (BD FACSCalibur, Becton Dickinson, San Jose, CA, USA). A total of 50,000 events were acquired for each sample and the percentage of cells in the different phases of the cell cycle was analyzed with BD CellQuest Pro 5.1 software (Becton Dickinson, San Jose, CA, USA).

### 4.7. Apoptosis Assay

To quantify the proportion of early-stage apoptotic cells the commercial kit “Vybrant Apoptosis Assay Kit #4” with YO-PRO-1 and PI for Flow Cytometry (Invitrogen, Thermo Fisher Scientific, Waltham, MA, USA; catalog number: V13243) was used. The cells were stained according to the supplemented manufacturer protocol. After 24 hours post-radiation, cells were collected and washed in cold phosphate-buffered saline (PBS). A total of 1 μL of YO-PRO-1 stock solution and 1 μL of PI stock solution were added to each 1 mL containing 1 × 10^6^ cells. Cells were incubated on ice for 20–30 min. Cells were analyzed by flow cytometry (BD FACSCalibur, Becton Dickinson, San Jose, CA, USA) using 488 nm excitation with green fluorescence emission for YO-PRO-1 (i.e., 530/30 bandpass) and red fluorescence emission for PI (i.e., 610/20 bandpass). A total of 50,000 events were acquired for each sample and analyzed with BD CellQuest Pro 5.1 software (Becton Dickinson, San Jose, CA, USA).

### 4.8. Analysis of ABCG2 Expression

Cells were seeded at concentrations of 2 × 10^3^ cells/0.32 cm^2^ into a 96-well plate. Cells were fixed in 4% PFA for 15 min at room temperature, followed by two rinses in PBS (pH 7.4) and permeabilization for 40 min with 0.3% Triton-X100 (in PBS, pH 7.4), supplemented by 2% BSA to block non-specific antibody binding. The fixed/permeabilized cells were then incubated for 1 h at room temperature in the dark with primary mouse anti-ABCG2 antibody diluted in PBS with 1% BSA and 0.3% Triton-X100 at 1:300. Following several rinses with PBS/0.05% Tween-20, cells were incubated with secondary goat anti-mouse IgG and IgM antibody conjugated to horseradish peroxidase (HRP) diluted in PBS with 1% BSA and 0.3% Triton-X100 at 1:10,000 for 1 h at room temperature in the dark. Cells were then rinsed three to four times with PBS/0.05% Tween-20. After that, cells were incubated at room temperature with premixed QuantaBlu Fluorogenic Peroxidase Substrate (Thermo Scientific, Thermo Fisher Scientific, Waltham, MA, USA) until the desired signal was produced. After attaining the desired signal, the Stop Solution (2N sulfuric acid) (Kirkegaard and Perry Laboratories Inc., Gaithersburg, MD, USA), was added to each well for 10 minutes to terminate substrate/peroxidase reaction and enhance sensitivity 2–4-fold. The fluorescent signal was measured at excitation/emission maxima of 325/420 (range of 315–345/370–460) using the CLARIOstar microplate reader (BMG LABTECH, Ortenberg, Germany), and MARS Data Analysis Software (BMG LABTECH, Ortenberg, Germany) was used to analyze the data obtained.

### 4.9. Statistical Analysis

Statistical and mathematical analyses of the data were conducted using the Statistica 8.0 software (StatSoft, Tulsa, OK, USA) and EXCEL 2010 Software (Microsoft, Redmond, WA, USA). The results are presented as means of three independent experiments ± standard error. Statistical significance was tested using Student’s *t*-test and the Mann–Whitney *U* test.

## Figures and Tables

**Figure 1 ijms-21-03342-f001:**
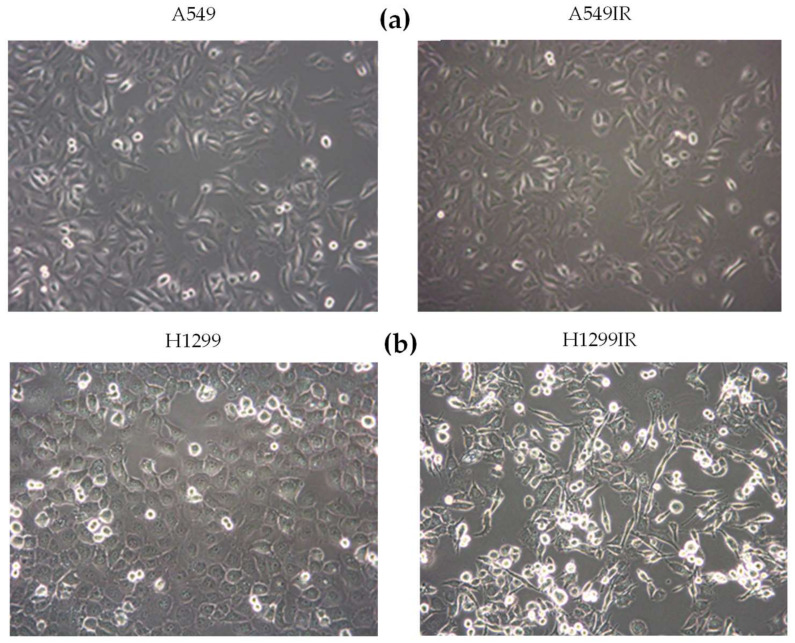
Phase contrast microscopic images of parental and irradiation-surviving A549 (**a**) and H1299 (**b**) cells. Objective: 20x.

**Figure 2 ijms-21-03342-f002:**
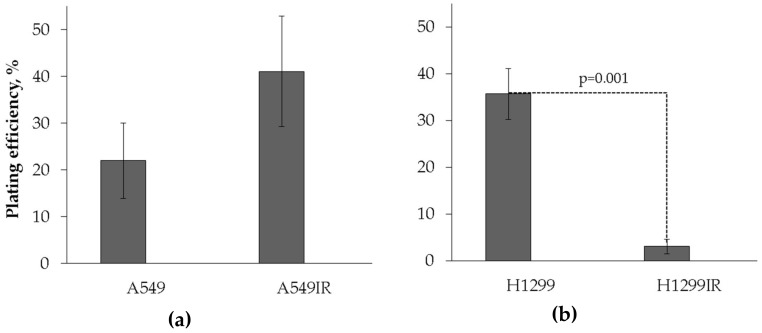
Plating efficiency of parental and irradiation-surviving A549 (**a**) and H1299 (**b**) cells. Colonies of the small size were counted under light microscope. Colonies ≥ 50 cells were counted. Data are means ± SEM of more than three independent experiments.

**Figure 3 ijms-21-03342-f003:**
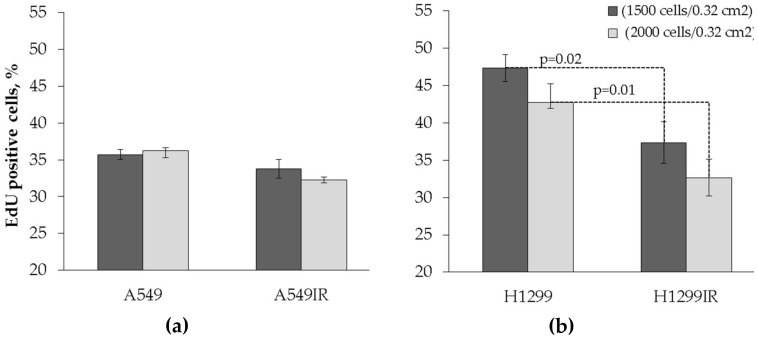
Assessment of the proliferative activity in both parental (non-irradiated) and irradiation-surviving A549 and H1299 cells using the Click-iT^TM^ EdU test (cells were seeded in 96-well plates at concentrations of 1500 and 2000 cells/0.32 cm^2^, marked by black and grey columns, respectively). (**a**) Changes in the percentage of EdU-positive cells in A549 and A549IR cell populations. (**b**) Changes in the percentage of EdU-positive cells in H1299 and H1299IR cell populations. Cell counting was performed at objective 10×. Data are means ± SEM of more than three independent experiments.

**Figure 4 ijms-21-03342-f004:**
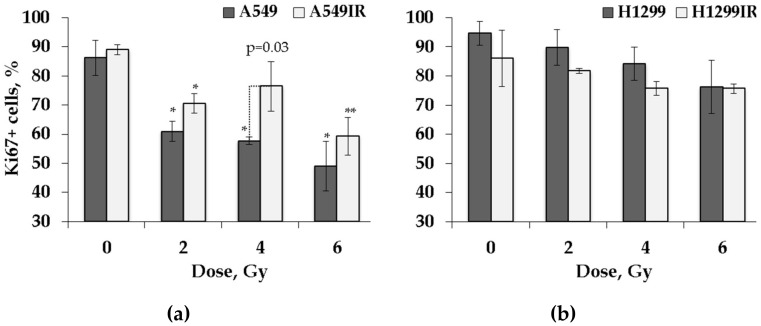
Proliferation activity of parental and irradiation-surviving non-small cell lung cancer (NSCLC) cells 24 h after exposure to different doses of X-rays. Changes in proliferation activity of A549 and A549IR cells (**a**) and H1299 and H1299IR cells (**b**) were analyzed 24 h after exposure to 2, 4, and 6 Gy of X-rays. * denotes significant differences between groups at *p* < 0.05. ** denotes significant differences between groups at *p* < 0.001. Data are means ± SEM of more than three independent experiments.

**Figure 5 ijms-21-03342-f005:**
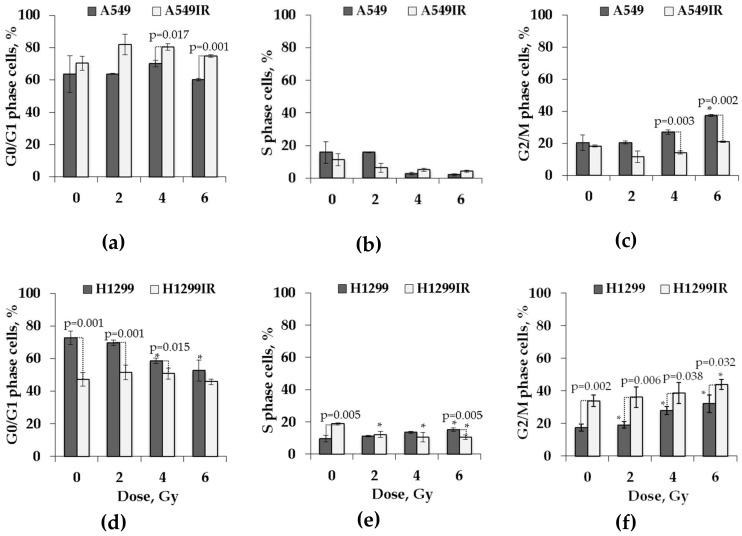
Cell cycle analysis of parental (A549 and H1299) and irradiation-surviving (A549IR and H1299IR) NSCLC cells 24 h after exposure to different single doses of X-rays. Proportion of A549 and A549IR cells in G0/G1 (**a**), S (**b**), and G2/M (**c**) cell cycle stages. Proportion of H1299 and H1299IR cells in G0/G1 (**d**), S (**e**), and G2/M (**f**) cell cycle stages. * denotes significant differences compared to unirradiated cells at *p* < 0.05. A total of 50,000 events were acquired for each sample. Data are means ± SEM of more than three independent experiments.

**Figure 6 ijms-21-03342-f006:**
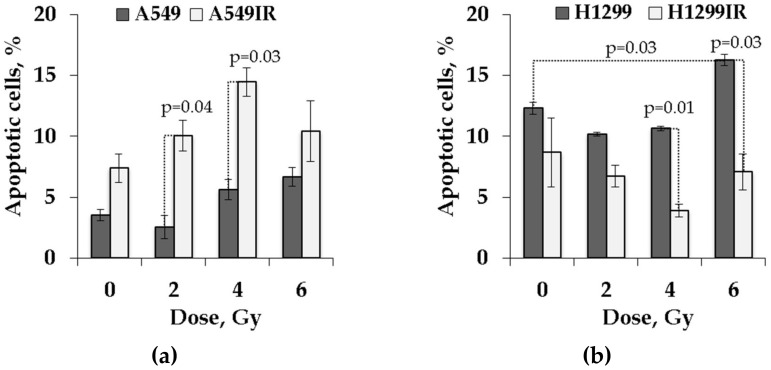
Assessment of apoptosis by YO-PRO-1 in parental and irradiation-surviving NSCLC cell lines. (**a**) A549IR cells demonstrated a higher rate of apoptotic (YO-PRO-1-positive/propidium iodide (PI)-negative) cells 24 h after exposure to IR compared to parental A549 cells; (**b**) H1299IR cells showed lower percentage of apoptotic (YO-PRO-1 positive/PI negative) cells after IR exposure compared to parental H1299 cells. A total of 50,000 events were acquired for each sample. Data are means ± SEM of more than three independent experiments.

**Figure 7 ijms-21-03342-f007:**
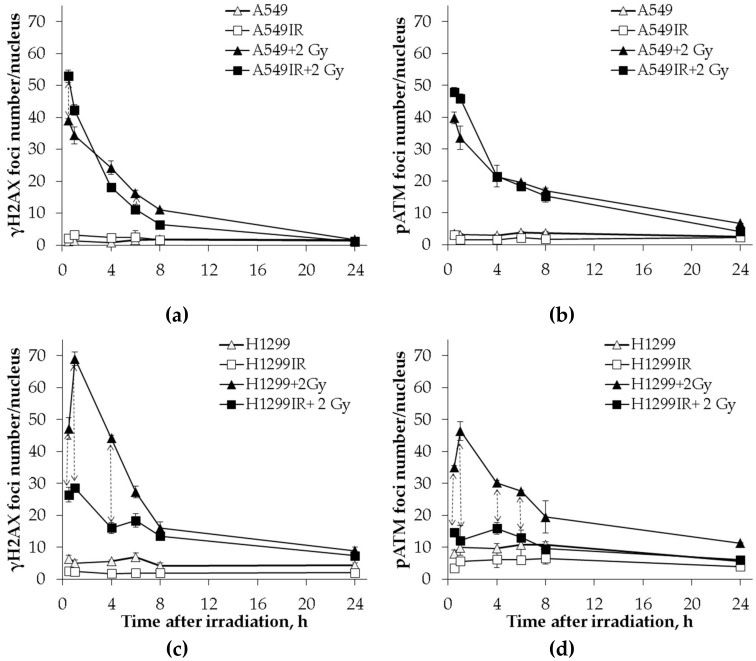
Kinetics of γH2AX and phosphorylated ataxia telangiectasia mutated (pATM) foci changes in A549 and A549IR cells and H1299 and H1299IR cells after exposure to an extra single dose of 2 Gy of X-rays. Comparative analysis of changes in the number of γH2AX foci in A549 and A549IR (**a**) and H1299 and H1299IR cells (**c**) after 2 Gy X-ray exposure; changes in the number of рАТМ foci in A549 and A549IR cells (**b**) and H1299 and H1299IR cells (**d**) after 2 Gy X-ray exposure. ↕ denotes significant differences between groups at *p* < 0.05. Data are means ± SEM of more than three independent experiments.

**Figure 8 ijms-21-03342-f008:**
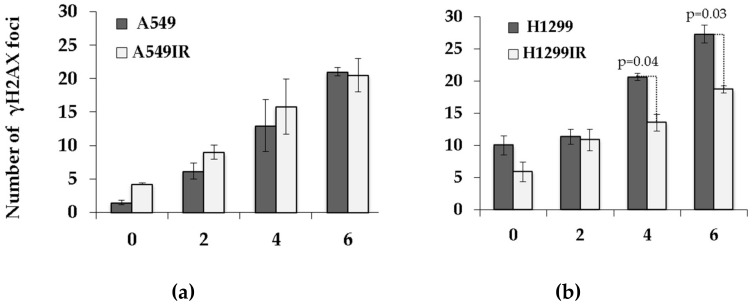
Changes in the number of residual γH2AX, 53BP1 foci, and their co-localization was analyzed in A549 and A549IR cells and H1299 and H1299IR cells 24 h after exposure to different single doses of X-rays. The number of residual γH2AX foci increased post-IR at extra single doses in A549 and A549IR cells (**a**) and H1299 and H1299IR cells (**b**). Changes in the number of residual 53BP1 in A549 and A549IR cells (**c**) and H1299 and H1299IR cells (**d**). Changes in the number of co-localized γH2AX/53BP1 in A549 and A549IR cells (**e**) and H1299 and H1299IR cells (**f**). Data are means ± SEM of more than three independent experiments.

**Figure 9 ijms-21-03342-f009:**
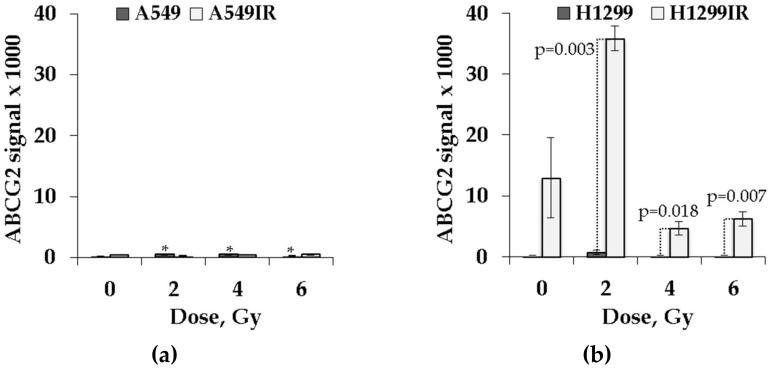
ABCG2 expression profile in parental and irradiation-surviving NSCLC cell lines. (**a**) A549IR cells did not show any difference in ABCG2 expression 24 h after exposure to IR compared to parental A549 cells; (**b**) H1299IR cells showed statistically significantly higher level of ABCG2 expression after IR exposure compared to parental H1299 cells. ABCG2 signal normalized to the number of cells. * denotes significant differences between correspondent control groups at *p* < 0.05. Data are means ± SEM of more than three independent experiments.

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
