# Peer review of "The p53–53BP1-Related Survival of A549 and H1299 Human Lung Cancer Cells after Multifractionated Radiotherapy Demonstrated Different Response to Additional Acute X-ray Exposure"

_ijms, 2020, doi:10.3390/ijms21093342_

Round 1
Reviewer 1 Report
This manuscript shows the different cell responses of A549 (p53 wild-type) and H1299 (p53 null) that survived after multifractionated irradiations. The present results are very interesting and the story of this study is well constructed. In particular, the authors focused on p53 status and cancer stem cells. There are a few major comments and several specific comments before going to publication.
General comments:
- I believe that p53 status and stem-like nature are important factors for determining radioresistant degree for human non-small cell lung cancer (NSCLC) based on the series of the experimental results. However, there is no term of “p53 status” and “cancer stem cell” in the title. Why did not the author describe the terms in the title? There is no strong scientific evidence here? Please explain. I believe that addition of these to the title can bring the impact in the field of radiation therapy.
- The authors stated what they experimentally did in this work in abstract section. This description seems to show less impact in the field of radiotherapy and radiation biology. The authors should revise the last sentence in the abstract section (lines 33-35). I believe the description how the preset experimental data can contribute to clinical outcome (e.g., cell killing and tumor control probability) should be added.
- This study shows clonigenic survival (proliferation) and apoptosis (DNA damage and repair process). According to the recent reports, inflammatory response (NF-kB and COX2 pathways) is also essential factor for determining the cell fate (survival and apoptosis) and for inducing radio-resistance after irradiation (Hamada et al., Current Molecular Pharmacology (2011) 4, 79-95; Steve et al., Mol Cancer Res (2008) 6(1); Michael et al., J Mammary Gland Biol Neoplasia (2006) 11, 63–73; Guozheng et al., Oncogene (2004) 23, 535–545). Did the authors previously evaluate the relation between this study and inflammatory responses? At least, please add discussion for the role of inflammatory responses inducing radio-resistance.
- I agreed well with the idea that p53 status plays a key role for discussing the different cell responses of A549 and H1299 (lines 167-169, 223-224, 270-275 and so on). Intrinsically, is there any possibility that the p53 status differs between parent cells and IR cells? Did you measure the level of p53 activation after irradiation?
Specific comments:
- Introduction (lines 69): I believe the phrase should be change “The aim of the present study was to establish and characterize …” into “The aim of the present study is to investigate the nature of NSCLC cell lines (p53 wild-type A549 and p53-null H1299) survived after multi-fractionated irradiations in total dose of 60 Gy”.
- Introduction (lines 71-73): The phrase “A549 cells have a functional…” should be moved to the material and method section. Instead of that, the authors should better to add a sentence what was found throughout this work.
- Results (line 115): What does the phrase “For this purpose” mean?
- Results (line 130): Please revise to be “radiation-survived”.
- Results (lines 146-147): “Non-irradiated A549IR and …” is incomplete sentence.
- Results (line 160): Please change “seen” to “observed”.
- Results (line 235): Please change “crucial” to “lethal”. And please change “introduced by” into “induced after”.
- Results (line 249): Please revise “0,5” to be “0.5”.
- Results (line 328): Please change “As shown on Fig. 9” to “As shown in Fig. 9”.
- Discussion (line 363): Please revise “P53” to “p53”.
- Discussion (line 386): What dose the phrase “constant irradiation” mean? Constant dairy dose per fraction?
- Figure legend: What is SE? It means standard deviation (s.d.)?
Author Response
Dear Reviewer,
We appreciate your time and effort in reading our manuscript and indicating potential deficiencies in our work and ways to improve it. Please find below our detailed responses to your comments.
General comments:
- I believe that p53 status and stem-like nature are important factors for determining radioresistant degree for human non-small cell lung cancer (NSCLC) based on the series of the experimental results. However, there is no term of “p53 status” and “cancer stem cell” in the title. Why did not the author describe the terms in the title? There is no strong scientific evidence here? Please explain. I believe that addition of these to the title can bring the impact in the field of radiation therapy.
4. I agreed well with the idea that p53 status plays a key role for discussing the different cell responses of A549 and H1299 (lines 167-169, 223-224, 270-275 and so on). Intrinsically, is there any possibility that the p53 status differs between parent cells and IR cells? Did you measure the level of p53 activation after irradiation?
Answer to related questions 1 and 4. : We agree and very much delighted with the Reviewer's suggestion to add the terms “p53 status” and “cancer stem cell” in the title. The reason why we did not include the “p53 status” before: although we considered the p53 status as an important and valuable parameter for both radioresistance and CSC in our studies, we felt that we did not make sufficient in-depth investigation and proof in current study leaving this point to ongoing study for the next our paper. Answering the Reviewer's fundamental question regarding the level of p53 activation after IR, indeed, we investigated not only the level of p53 expression but rather p53-family proteins (TAp73/ΔNp73 and p63), as H1299 having a biallelic deletion of the TP53 gene have no p53, as you know. The outcome results expected to come soon, although our current isolation due to COVID significantly delayed such essential data to be included in the current paper.
Regarding “cancer stem cell”, we felt that evident CSC-like morphology (spindle-like, loss of adhesion to the substrate, significant increase of viable round-shaped cells and decline in intercellular contacts) together with significantly higher ABCG2 expression observed in H1299IR cells compared to parental and both A549 and A549IR cells, can not be the sufficient proof that they are true CSC. That's why we carefully mentioned that they are of “CSC-like phenotype” in the paper. At the same time, we are now analyzing the expression of EMT signatures (N-/E-Cadherins vs. Vimentin/α-SMA) along with CSC putative markers (e.g., EpCAM/ CD133/CD166/CD44) in these cells to get more proof on that matter to be more confident in our conclusions in our next paper. Unexpected isolation of our researchers also delayed such data for the present paper.
Therefore, taking into account the Reviewer's kind suggestion and all mentioned above, we would carefully change the title to “The p53-53BP1-related survival of A549 and H1299 human lung cancer cells after multifractionated radiotherapy demonstrated different response to additional acute X-rays exposure.” We will also appreciate the Reviewer's opinion on this matter.
- The authors stated what they experimentally did in this work in abstract section. This description seems to show less impact in the field of radiotherapy and radiation biology. The authors should revise the last sentence in the abstract section (lines 33-35). I believe the description how the preset experimental data can contribute to clinical outcome (e.g., cell killing and tumor control probability) should be added.
Answer to question 2:
We highly appreciate and respect of the Reviewer's suggestions, and on 33-35 lines will include the following sentences: "Our study provides strong evidence that different DNA repair mechanisms are activated by MFR, as well as single dose IR, and that the enhanced cellular survival after MFR is reliant on both p53 and 53BP1 signaling along with NHEJ. Our results are of clinical significance as they can guide the choice of the most effective IR regimen by analyzing the expression status of the p53-53BP1 pathway in tumors and thereby maximize therapeutic benefits for the patients while minimizing collateral damages to normal tissue."
- This study shows clonigenic survival (proliferation) and apoptosis (DNA damage and repair process). According to the recent reports, inflammatory response (NF-kB and COX2 pathways) is also essential factor for determining the cell fate (survival and apoptosis) and for inducing radio-resistance after irradiation (Hamada et al., Current Molecular Pharmacology(2011) 4, 79-95; Steve et al., Mol Cancer Res (2008) 6(1); Michael et al., J Mammary Gland Biol Neoplasia (2006) 11, 63–73; Guozheng et al., Oncogene (2004) 23, 535–545). Did the authors previously evaluate the relation between this study and inflammatory responses? At least, please add discussion for the role of inflammatory responses inducing radio-resistance.
Answer to question 3:
Again, we much value at this point highlighted by the Reviewer. Indeed, we felt we did not get all convincing evidence (data) to mention this point in the current paper. Nevertheless, one of the main topics of our ongoing follow-up study is the investigation of Irf-NF-kB pathway in the development of radioresistance. After that, we will be glad to discuss this issue in-depth using convincing data obtained.
Specific comments:
- Introduction (lines 69): I believe the phrase should be change “The aim of the present study was to establish and characterize …” into “The aim of the present study is to investigate the nature of NSCLC cell lines (p53 wild-type A549 and p53-null H1299) survived after multi-fractionated irradiations in total dose of 60 Gy”.
Answer to specific comment 1: We highly appreciate and agree with Reviewer suggestion and made the respective change.
- Introduction (lines 71-73): The phrase “A549 cells have a functional…” should be moved to the material and method section. Instead of that, the authors should better to add a sentence what was found throughout this work.
Answer to specific comment 2: We agree with Reviewer comment and introduced the following: “ In the present study, we found that the H1299IR (p53-deficient) cells demonstrated a more prominent CSC-like phenotype in comparison to A549IR (p53-wild type) cells, and also suggesting that, while the proliferation of parental cancer cells is p53-dependent, the proliferation of the cells survived after MF IR exposure is p53-independent. Interestingly, in response to incrementing acute single dose IR exposure, functional p53 confer significant dose-dependent decrease in the fraction of Ki67+ to both A549 and A549IR cells, whereas in the absence of functional p53, both parental and MFR-survived cells acquired only subtle cell cycling decrease. Our data strongly support the notion that the presence of functional p53 resulted in durable IR-induced G1 and G2 arrests, whereas the absence of p53 leads both parental and IR-resistant NSLC cells to exit of G0/G1 and more prominent arrest in G2/M phase. Besides, our data draw attention to the efficient and p53-independent 53BP1 recruitment to damaged chromatin and 53BP1-mediated NHEJ, thus, favoring either the survival advantage p53-null cells (higher proliferation/DNA replication and lower apoptosis) or induction of CSC-like phenotype observed in the present study. Though, our current study demonstrated that 53BP1-related survival NSLC cells after MFR is not associated with p53 functionality. Moreover, in our experiments, the enhanced 53BP1-mediated cell survival after MFR does not point to any particular type of cell cycle checkpoint response and can not be amply explained by its role in regulating checkpoint or pro-apoptotic signaling. Present data add the value to debated p53-MDR relationship in cancer by indicating that p53 status affects ABCG2 expression in cells survived after MFR and indicate a major role of p53-family proteins in conferring a stem-like cell phenotype and radioresistance to NSCLC cells that are associated with ABCG2 overexpression.”
- Results (line 115): What does the phrase “For this purpose” mean?
Answer to specific comment 3: We highly appreciate the Reviewer’s comment on this sentence. The phrase “For this purpose” can be referred to the previous sentence: “Next, we investigated the impact of p53 on the proliferation of the sublines survived after multifraction X-ray irradiation“. The methods described in the sentence were used for the proliferation analysis. The specific changes were made.
- Results (line 130): Please revise to be “radiation-survived”.
Answer to specific comment 4: We highly appreciate the Reviewer’s suggestion and made the respective change.
- Results (lines 146-147): “Non-irradiated A549IR and …” is incomplete sentence.
Answer to specific comment 5: We highly appreciate and agree with Reviewer’s suggestion and made the respective change as “Non-irradiated A549IR and H1299IR as well as their parental cells were used as controls.”
- Results (line 160): Please change “seen” to “observed”.
Answer to specific comment 6: We appreciate the Reviewer ‘s comment. The respective change was made.
- Results (line 235): Please change “crucial” to “lethal”. And please change “introduced by” into “induced after”.
Answer to specific comment 7: We agree with Reviewer ‘s comment and made the respective change.
- Results (line 249): Please revise “0,5” to be “0.5”.
Answer to specific comment 8: Corrected.
- Results (line 328): Please change “As shown on Fig. 9” to “As shown in Fig. 9”.
Answer to specific comment 9: We highly appreciate the Reviewer ‘s suggestion and made the respective change.
- Discussion (line 363): Please revise “P53” to “p53”.
Answer to specific comment 10: We highly appreciate and agree with Reviewer ‘s suggestion and made the respective change.
- Discussion (line 386): What dose the phrase “constant irradiation” mean? Constant dairy dose per fraction?
Answer to specific comment 11: We apologize that the sentence is not clear. As the Reviewer noticed the “constant irradiation” term here refers to multifraction radiotherapy (or constant daily dose per fraction, as the Reviewer noticed). According to the Reviewer ‘s comment, corresponding changes were introduced.
- Figure legend: What is SE? It means standard deviation (s.d.)?
Answer to specific comment 12: We apologize for the incorrect presentation of our results. The “SE” refers to “standard error”. Corresponding changes were introduced in the description of Fig.2-Fig.9.
Reviewer 2 Report
The work by Postovalova and co-workers is an experimental in vitro study of NSCLC surviving a total dose of 60 Gy given in fractions of 2, 5, and 10 Gy. The response to further irradiation with respect to H2AX, ATM, and 53Bp1phospyrylation and foci kinetics as well as proliferation, cell cycle, cell death and ABCG2 protein expression was studies. The study of such a selected subpopulation is interesting and relevant to understand treatment responses in patients. The experiments appear to be well planned and performed, however, the interpretation of some of the data is questionable (see below). The manuscript is overall well written but the results section is too long and some of the material should be moved to the discussion or the material and method (for example lines 115-129). In addition, the manuscript should be edited for language improvement. Examples: the use of “survived” cells (either cells that survived or surviving cells) or the sentences lines 130-132, lines 151-154.
Major concerns:
- How was the fractionation scheme chosen? Is the combination of 10 fractions of 2 Gy, 4 fractions of 5 Gy and 2 fractions of 10 Gy used in the clinic? Isn’t 2 times 10 Gy used for pallative treatment? I would think it would more interesting and relevant to compare fractionations with 2 Gy or 10 Gy.
- Line 87. It is not evident that the described morphological changes are related to EMT.
- Line 144: Ki67 can not be used to measure clonogenic survival, only the number of cells that are proliferating. The study would greatly benefit from including clonogenic survival data (obtained as in figure 2) with the doses used for the other assays.
- Lines 127- 138: As the cells were only exposed to EdU for 2.5 h (methods), this assay will only measure the amount of cells that were in S-phase at that time, not the total proliferating population.
- Lines 177-200. I think it would be more relevant to look at the cell cycle as a whole for each cell population, not each phase separately. A G1 or G2 arrest will influence (and explain) the relative amount of cells in the other phases.
- Lines 215-228: As rightly stated in line 214, a marker for compromised cell membrane will be an early marker of cell death. However, necrosis is related to a compromised plasma membrane, not apoptosis. On the contrary, apoptosis is characterized by membrane blebbing without leakage.
Author Response
Dear Reviewer,
We appreciate your time and effort in reading our manuscript and indicating potential deficiencies in our work and ways to improve it. Please find below our detailed responses to your comments.
Major concerns:
- How was the fractionation scheme chosen? Is the combination of 10 fractions of 2 Gy, 4 fractions of 5 Gy and 2 fractions of 10 Gy used in the clinic? Isn’t 2 times 10 Gy used for pallative treatment? I would think it would more interesting and relevant to compare fractionations with 2 Gy or 10 Gy.
Answer: We agree with the Reviewer's comment as this question was not properly introduced in the manuscript.
Introduction will include the following sentences: "For locally advanced NSCLC the standard treatment approach is conventionally fractionated (1.8–2.0 Gy/day) radiotherapy to a dose of approximately 60–66 Gy with concurrent, platinum-based chemotherapy (McDonald and Popat 2014, Falkson, Vella et al. 2017). Stereotactic Body Radiation Therapy (SBRT) is a potentially effective technology to manage early-stage NSCLC for inoperable patients and is often used for lung cancer patients with thoracic recurrences previously treated with conventional radiotherapy (Milano, Kong et al. 2019). SBRT allows clinicians to administer high doses (commonly used doses are 48–50 Gy in 4–5 fractions and 50–60 Gy in 8–10 fractions) to accurately target the tumor (Prezzano, Ma et al. 2019). However, both local and regional/distant recurrences still occur after SBRT (Kumar and McGarry 2019). In this study we applied multiple fractions of subsequently increasing doses to obtain the populations of radioresistant cells. The combination of 10 fractions of 2 Gy, 4 fractions of 5 Gy and 2 fractions of 10 Gy is not used in the clinic. However, this combination could serve as the representation of the conventional radiotherapy treatment with subsequent SBRT".
Fractionation scheme of 2 times 10 Gy is not used for palliative treatment, rather 2 fractions 8 Gy or 1 fraction 10 Gy. However, even during the course of palliative RT, RT fractionation schemes vary and are not always well-supported by randomized data. There is no strong evidence that any regimen gives greater palliation. An exploratory meta-analysis suggested that more aggressive radiation schedules (greater than 30 Gy/10 fractions) were associated with modest improvements in survival (5-6% at 1 year and 3% at 2 years), 21 though with greater acute side effects, such as radiation esophagitis (F Macbeth, 2001 https://doi.org/10.1002/14651858.CD002143; Jason F Lester, 2006 https://doi.org/10.1002/14651858.CD002143.pub2). In this case various shorter EBRT dose/fractionation schedules (eg, 20 Gy in 5 fractions, 17 Gy in 2 weekly fractions, 10 Gy in 1 fraction), which provide good symptomatic relief with fewer side effects, can be used for patients requesting a shorter treatment course and/or in those with a poor performance status (George Rodrigues et al. 2011 doi: 10.1016/j.prro.2011.01.005). Multiple trials showed no difference in pain control between single fraction and longer radiation courses for patients with bone metastases. Although there are potential clinical scenarios when extended fractionation may be warranted, in most palliative settings, radiation treatments can safely and effectively be given in ≤10 fractions (Miranda B. Lam et al.2018).
The effects of 2 Gy fractionations in human NSCLC cell lines are well described in the literature (doi:10.1158/1535-7163.MCT-13-0608;DOI:10.177/1010428317695010; https://doi.org/10.1080/15384047.2016.1139232; https://doi.org/10.1016/j.athoracsur.2011.07.032; https://doi.org/10.5808/GI.2013.11.4.245). The usage of 10 Gy fractionations is less common (DOI: https://doi.org/10.1016/j.ijrobp.2004.07.182). We previously failed to establish radioresistant NSCLC cells using 6x10 Gy fractionation scheme with irradiation once a week. The majority of cells died after the second exposure. We believe that cells need longer periods between fractions to recover. The protocol will therefore undergo further optimization and 10 Gy fractionation scheme will be used in our next study for the establishment of radioresistant NSCLC cells. The literature also include the establishment of radioresistant NSCLC cell using 5x6 Gy (https://doi.org/10.3892/ol.2019.9888) and 20x4 Gy (https://doi.org/10.1371/journal.pone.0175977) irradiation schemes.
- Line 87. It is not evident that the described morphological changes are related to EMT.
Answer: We partially agree with the Reviewer's comment. Indeed, our light microscopy images showed disruption of the cell to cell junctions, loss of epithelial characteristics and distinct change in morphology (toward spindle-like, loss of adhesion to the substrate, significant increase of viable round-shaped cells) in the IR cells as shown on Fig.1. We believed that such morphological changes are related to EMT, as published in many papers [reviewed in e.g. "Epithelial-Mesenchymal Plasticity: A Central Regulator of Cancer Progression.Ye X, Weinberg RA. Trends Cell Biol. 2015 Nov; 25(11):675-686]. Although we agree with Reviewer's comment and aware that we did not support this with data on invasiveness, motility, and the expression of EMT signatures (N-/E-Cadherins vs. Vimentin/α-SMA), which is the subject of ongoing studies. Truly, we originally carefully named such morphology as "… prominent epithelial-mesenchymal transition (EMT)-like phenotype …", but due to typing mistake, it was wrongly written as "...prominent epithelial-mesenchymal transition (EMT) phenotype…". We appreciate the Reviewer's comment pointing out this discrepancy, and corrected this mistake.
- Line 144: Ki67 can not be used to measure clonogenic survival, only the number of cells that are proliferating. The study would greatly benefit from including clonogenic survival data (obtained as in figure 2) with the doses used for the other assays.
Answer: We agree with the Reviewer's comment regarding Ki67 and clonogenic survival. Indeed, in our paper, we were going only "To examine the relationship between DNA replicating response and clonogenic survival after fractionated IR exposure…". We also respect Reviewer's comment regarding the beneficial role of clonogenic survival data. Though, our current unexpected isolation due to COVID significantly delayed such essential data to be included in the current paper in time. In the current situation, we would propose to leave these experiments for our next paper.
- Lines 127- 138: As the cells were only exposed to EdU for 2.5 h (methods), this assay will only measure the amount of cells that were in S-phase at that time, not the total proliferating population.
Answer: We agree with Reviewer comment that this method only measure the percentage of S-phase cells in the population, not the total proliferating population. The recommended by the manufacturer's starting concentration of Edu is 10 μM for 1–2 hours. For longer incubations, recommended using lower concentrations. Altering the amount of time the cells are exposed to EdU or subjecting the cells to pulse labeling with EdU allows the evaluation of various DNA synthesis and proliferation parameters. Sufficient time intervals for pulse labeling and the length of each pulse depend on the cell growth rate.
Our preliminary optimization experiments demonstrated the best conditions we described in the Materials and Methods, taking into account the proliferation rate of our cell lines and their IR sub-lines. Our methodology corroborates the one used in several publications on the same cell lines [e.g. Tang, Y., Cui, Y., Li, Z. et al. Radiation-induced miR-208a increases the proliferation and radioresistance by targeting p21 in human lung cancer cells. J Exp Clin Cancer Res 35, 7 (2016). https://doi.org/10.1186/s13046-016-0285-3; Zhao Z, Liu B, Sun J, et al. Scutellaria Flavonoids Effectively Inhibit the Malignant Phenotypes of Non-small Cell Lung Cancer in an Id1-dependent Manner. International Journal of Biological Sciences. 2019 ;15(7):1500-1513. DOI: 10.7150/ijbs.33146; Cao B, Tan S, Tang H, Chen Y, Shu P. miR‑512‑5p suppresses proliferation, migration and invasion, and induces apoptosis in non‑small cell lung cancer cells by targeting ETS1. Molecular Medicine Reports. 2019 May;19(5):3604-3614. DOI: 10.3892/mmr.2019.10022.]
In general, Click-iT® EdU method - the most accurate method of directly measuring DNA synthesis. EdU (5-ethynyl-2´-deoxyuridine) is a nucleoside analog to thymidine and is incorporated into DNA during active DNA synthesis. Standard flow cytometry methods are used for determining the percentage of S-phase cells in the population. In our present study, we used Ki67 staining as a supplementary proliferation-related method, to quantify cycling cells, i.e., the amount of DNA replicating cells and the cells in the growth – pre-replicative phase ( as mentioned on lines 151 and 168), to complete our observations and conclusions regarding the proliferation of cells.
- Lines 177-200. I think it would be more relevant to look at the cell cycle as a whole for each cell population, not each phase separately. A G1 or G2 arrest will influence (and explain) the relative amount of cells in the other phases.
Answer: We are very appreciated your suggestion but pictures made in this way are looked very overloaded.
- Lines 215-228: As rightly stated in line 214, a marker for compromised cell membrane will be an early marker of cell death. However, necrosis is related to a compromised plasma membrane, not apoptosis. On the contrary, apoptosis is characterized by membrane
Answer: Indeed, until recently, necrosis was thought to be an unregulated process. However, there are two broad pathways in which necrosis may occur in an organism: The first one initially involves oncosis, where swelling of the cells occurs. Affected cells then proceed to blebbing, and this is followed by pyknosis, in which nuclear shrinkage transpires. In the final step of this pathway cell nuclei are dissolved into the cytoplasm, which is referred to as karyolysis. A second pathway is a secondary form of necrosis that is shown to occur after apoptosis and budding. In these cellular changes of necrosis, the nucleus breaks into fragments (known as karyorrhexis). None of these features observed in our experiments after prolonged or acute IR of cells. Besides, necrosis, if any, occurs at much later time points then 24 hours after treatment used in our study. Altogether, we do not believe that the necrosis will have a significant contribution to the apoptotic-related changes in membrane integrity we observed and measured in the present study.
Round 2
Reviewer 2 Report
The manuscript is improved but there still remain some issues
Line 188: You should not use the term clonogenic survival. Ki67 is a marker for proliferation, but it does not measure clonogenic survival.
Lines 215-228: The problem is not the pathway of necrosis. The problem is that a leaky membrane is not associated with apoptosis. For example: A standard assay for measurement of apoptosis is using an antibody for annexinV combined with PI to gate the cells with a leaky membrane, which is a marker for necrosis.
Author Response
We are grateful to the Reviewer for these comments.
Comment 1. Line 188: You should not use the term clonogenic survival. Ki67 is a marker for proliferation, but it does not measure clonogenic survival.
Answer to comment 1: We highly appreciate the Reviewer’s comment and apologize for the previous misunderstanding. The Ki67 indeed was not used in our study to measure the clonogenic survival. We intended only to measure the proliferation activity of both parental and IR-survived sublines after single-dose X-ray exposure. The respective change made in the text.
Comment 2: Lines 215-228: The problem is not the pathway of necrosis. The problem is that a leaky membrane is not associated with apoptosis. For example: A standard assay for measurement of apoptosis is using an antibody for annexinV combined with PI to gate the cells with a leaky membrane, which is a marker for necrosis.
Answer to comment 2:
We apologize for the improper explanation. The term the leakiness of membrane usually regarded as either passive transport through the damaged membrane (which is happening during necrosis) or active transport through activated channels and/or receptors (mainly happening during early stages of apoptosis). Apoptosis is distinguished from necrosis, or accidental cell death, by characteristic morphological and biochemical changes, including compaction and fragmentation of the nuclear chromatin, shrinkage of the cytoplasm (due to membrane leakiness to e.g., water and K+ ions through activated Aquaporins and Voltage-gated plasma membrane potassium channels, respectively) and loss of membrane asymmetry. The later process is due to exposure of phosphatidylserine (PS) on the outer leaflet of the plasma membrane, which is a surface change common to many apoptotic cells. In this regard, we agree with the Reviewer's comment that Annexin V detects the appearance of PS residues caused by scramblases, which are activated by caspase cleavage in apoptotic cells. In this assay, the PI used to define necrotic cells that have broken plasma and nuclear membranes. Vital, certain dyes, such as the green fluorescent YO-PRO®-1 dye can enter only apoptotic cells (due to active transport, see below), whereas other dyes, such as the red fluorescent dye, propidium iodide (PI), cannot. Indeed, in our present study, we used both dyes to identify apoptotic and necrotic cells separately according to the commercial kit protocol (Vybrant™ Apoptosis Assay Kit (Invitrogen™, USA).
YO-PRO-1 is an early marker of apoptotic cells (reviewed in Wlodkowic et al., 2011, doi: 10.1016/B978-0-12-385493-3.00004-8). Its relatively large size (630 Da) prevents this dye from penetrating the intact plasma membrane of living cells. However, apoptotic processes jeopardize membrane integrity allowing YO-PRO-1 to enter the cells. The mechanism involved incorporates the release of ATP and UTP molecules into the extracellular space, leading to the activation of P2X7 receptors (Virginio et al. 1999). The opening of cation channels follows, allowing entry of YO-PRO-1 (Chekeni et al. 2010; Michel et al. 2000; Virginio et al. 1999). Functional activity of P2X7 receptors can influence apoptotic pathways (Chow et al. 1997), and activation of these receptors has been used to promote cell death in cancer cells (Gorodeski 2009). YO-PRO-1 positivity may be an early indicator of P2X7 receptor activation.
In our paper, we did not use the terms neither "leakiness" nor "leaky membranes" because of their broad interpretation, potentially leading to confusion. For better clarity and to avoid further misunderstanding, we modified our text as follows:
"To test this hypothesis, we investigated the effect of single-dose acute IR on radiation-induced apoptosis of parental and irradiation-survived cells using YO-PRO-1 and propidium iodide (PI) staining. The P2X7 receptor (P2X7R) (an adenosine triphosphate (ATP)-gated, non-selective cation channel) was shown to be activated during apoptosis leading to the conversion of a non-selective cation channel to a cytolytic pore (Kopp, Krautloher, et al. 2019). Certain dyes, such as the green fluorescent YO-PRO-1 dye, can enter these pores, whereas other dyes, such as PI, cannot. Thus, YO-PRO-1 can serve as an early marker of apoptotic cell death (Fujisawa, Romin, et al. 2014)"
We would also mention, as indicated in Figure 6 legend, that only YO-PRO-1 positive/PI negative cells were accounted as the apoptotic cells.
Round 3
Reviewer 2 Report
I approve of the changes in the manuscript and the answers to the review. However, I still have some issues.
Line 36: "Our study provides strong evidence that different DNA repair mechanisms are activated by multifraction radiotherapy (MFR), as well as single dose IR, ..." Should it be "compared to single dose IR"? There are no data from single dose IR in the study to compare with.
Line 38: "Our results are of clinical significance as they can guide the choice of the most effective IR regimen". How can they be used to guide the choice of regimen, when only one regimen has been investigated?
lines 99-100. " favoring either the survival advantage p53-null cells". I do not understand what is meant here.
Lines 111-112: "Exponentially growing A549 and H1299 cells were irradiated with clinically relevant doses of IR (10 fractions of 2 Gy, 4 fractions of 5 Gy and 2 fractions of 10 Gy)." It should be emphasized that all of these fractions are given together to the same cells, and that this is not cliniccaly relevant.
line 113: " the survived cells demonstrated enlarged flattened, morphology."
In the next paragraph this is described differently?
"A549IR cells did not differ in their morphology compared to parental A549 (p53-wild type) cells. The H1299IR cells morphologically changed into a spindled or rounded shape in contrast to
parental "
Line 325-330: The values of H2AX and pATM differs most 0.5-1 h post irradiation, which is too short for repair of DSBs. The amount of DSBs after 0.5 h should be approximately the same after a certain dose, since it is a result of the local energy deposition, not of biological processes. Therefore, it is more likely the detection and not the repair of DSBs that is affected.
line 432: "Clinically relevant mulitple fractions". See comment above and in earlier reviews.
line 629: It should be stated how the cells were counted before seeding for the colony assay.
I pointed out in the first round of review that the result section is very long and contains elements that belong in the discussion. This has not been addressed.